# Robot-Assisted Eccentric Contraction Training of the Tibialis Anterior Muscle Based on Position and Force Sensing

**DOI:** 10.3390/s19061288

**Published:** 2019-03-14

**Authors:** Keisuke Kubota, Masashi Sekiya, Toshiaki Tsuji

**Affiliations:** 1Graduate School of Saitama Prefectural University, Graduate Course of Health and Social Services, 820 Sannomiya, Koshigaya-Shi, Saitama 343-8540, Japan; 2Graduate School of Science and Engineering, Saitama University, 225 Shimo-Okubo, Sakura-Ku, Saitama-Shi, Saitama 338-8570, Japan; dendenkiki@gmail.com (M.S.); tsuji@mail.saitama-u.ac.jp (T.T.)

**Keywords:** position sensing, force sensing, rehabilitation, training robot

## Abstract

The purpose of this study was to determine the clinical effects of a training robot that induced eccentric tibialis anterior muscle contraction by controlling the strength and speed. The speed and the strength are controlled simultaneously by introducing robot training with two different feedbacks: velocity feedback in the robot controller and force bio-feedback based on force visualization. By performing quantitative eccentric contraction training, it is expected that the fall risk reduces owing to the improved muscle function. Evaluation of 11 elderly participants with months training period was conducted through a cross-over comparison test. The results of timed up and go (TUG) tests and 5 m walking tests were compared. The intergroup comparison was done using the Kruskal-Wallis test. The results of cross-over test indicated no significant difference between the 5-m walking time measured after the training and control phases. However, there was a trend toward improvement, and a significant difference was observed between the training and control phases in all subjects.

## 1. Introduction

Due to the extension of the average life expectancy and the decline in the birth rate, population aging is progressing on a global scale [1]. One of the countermeasures for this phenomenon is elongation of the healthy life expectancy of the elderly. Healthy life expectancy is defined as the average number of years that a person can expect to live in “full health” by taking into account years lived in less than full health due to disease and/or injury [2]. Increasing the healthy life expectancy leads to an increase in the number of self-sustaining elderly individuals.

Maintaining physical function and activity frequency is important in increasing healthy life expectancy, and securing mobility ability is crucial for this purpose. Since walking is the most common form of mobility, preventing falls during walking is of great importance. Therapy for walking ability improvement to prevent falls typically included balance training, joint range of motion (ROM) training, and muscle training. Many studies have reported that when muscle training is included in physical therapy programs for the elderly, the result is improvement in walking ability [3]. Furthermore, the previous study indicated that muscle training improves the ability to balance [4]. The function of muscle activity during walking has been clarified in many studies. The tibialis anterior muscle has an important function of body support in the early stance [5]. Previous studies have indicated that with aging, the muscle strength of the tibialis anterior muscle decreases [6], which is inevitable.

Tibialis anterior muscle activity during gait has an amplitude equivalent to 45% according to the MMT (manual muscle test) [7]. In other words, the tibialis anterior muscle during gait indicated activity of only about half of the maximal voluntary contractions. Moreover, in previous studies [4,8], walking-speed and functional performance related to walking ability were found to improve at low-intensity training. Many studies report that older adults increase muscle coactivation of the muscles around the ankle joint to avoid falls and posture instability [9,10]. The elevated co-contraction about the ankle in older adults is associated with increased fall risk [11,12]. In sum, the maintenance of the tibialis anterior muscle function is more important than increasing muscle strength. The previous study has specified that the tibialis anterior muscle decreases plantar flexion of the ankle by eccentric contraction at the early stance [7]. There are a variety of interventional methods used for improving the tibialis anterior muscle function. Relatively simple exercises are those that involve use of the TheraBand and other elastic bands [3]. These tools are reasonably effective in general, and they can be used only for training. On the other hand, the use of devices makes it possible to maintain a constant resistance force and angular velocity. Therefore, quantitative eccentric contraction training can be performed.

Currently, there exist several methods of eccentric contraction training of the tibialis anterior muscle using devices. Roy et al. developed a novel ankle robot called “anklebot” [13]. In addition, Yonezawa et al. are developing a Stewart platform-type ankle exercise device known as “parallel link type human ankle rehabilitation assistive device (PHARAD)” [14,15]. With regard to the point of eccentric contraction, the Biodex System4, produced by Biodex Medical Systems (Shirley, NY, USA), is widely used. Originally, it is quite difficult to adjust the load during eccentric contraction, since eccentric contraction training requires controlling both the speed and load simultaneously. Simultaneous control of force and speed is generally considered to be difficult in the field of robotics. With force control, providing appropriate load is possible; however, adjusting the speed becomes difficult. On the other hand, with position control, appropriate speeds can be achieved, whereas appropriate load cannot be provided. To overcome this problem, our group has developed a system that enables exercise with appropriate load and speed by providing the trainee with visual feedback [16]. Appropriate load and speed are simultaneously achieved by coordinated control, wherein the trainee controls the force while the robot performs position control. Although our previous study showed that the control system recognizes eccentric contraction with an appropriate load, no significant difference was observed in the timed up-and-go (TUG) score as the number of subjects was small and evaluation methods were limited [16]. As an extended work, this study determined the clinical effects by conducting a cross-over study with 11 subjects including a 5-m walking test for evaluation. The results and their corresponding discussion elucidate the training effect of the proposed training system.

## 2. Materials and Methods

### 2.1. Training Robot

The training robot is equipped with two DC motors. The motors are controlled by a PC, based on angle information acquired from optical encoders attached to the motors. Force sensors attached to the footplate enable the acquisition of force data. The data obtained at the tip of the footplate are visualized by computer graphics, and the trainee performs the eccentric contraction exercise by adjusting the manifested force while referring to this force data. Table 1 shows the specification of the training system.

Figure 1 displays an image of the training robot. The footplate of the robot can perform plantar flexion and dorsal extension of the ankle joint and up-and-down movement of the lower limbs by means of the parallel mechanism actuated by the two motors.

Figure 2 shows the control system of the training robot. The robot can control the angle and height from the ground to the footplate. It can be considered as a velocity control system with a PI controller. Table 2 shows parameters of the control system. *K_p_* and *K_d_* denote the position and velocity gains, respectively. θ denotes the angle of motors. *J*, *i*, *g*, and *n* denote inertia of moment, motor current, cutoff frequency of pseudo derivative, and reduction ratio of the gear, respectively. Res, cmd and ref in the subscripts denote response, command and reference values, respectively. $\tau_{ext}$ and *k_t_* denote external torque and torque constant, respectively. This study considered only the dorsiflexion and plantar flexion of the ankle joint; therefore, the angle of the rear motor controlling the ankle height remains constant, and a wave input is provided to the front motor controlling the tip height. With such a controller, dorsiflexion and plantar flexion of the ankle joint can be achieved while maintaining a constant heel position.

From control engineering studies, it is known that simultaneous position and force control cannot be achieved in robots [17]. This can also be assumed from the fact that the reaction force is uniquely determined by kinetics when the robot arm position is provided. In the case where the robot is in contact with an object, the reaction force is determined by the relative position of the object and the robot. In the case where the robot is in free motion without coming into contact with anything, the reaction force is zero.

However, eccentric contraction training requires appropriate force along with sufficient velocity, which is determined based on position control. In general, it is difficult to accomplish a position/velocity command and a force command simultaneously. Compliance control is a common solution for the system with position and force commands, despite resulting in compromising the two commands. This study addresses the issue by introducing a control architecture shown in Figure 3. The lower part of the block diagram stands for the control system described in Figure 2, which denotes the common position control of a robot. The upper part shows that the external force applied by the subject works as a kind of feedback with the subject adjusting the force based on the information on the display. External force applied from the subject becomes more accurate by substituting the somatic sense of the tip force to biofeedback with force visualization. This system can accomplish both position control and force control by individually distributing each role to the robot and the subject. 

### 2.2. Training Methods

As shown in Figure 4, during training, the trainee sits on a chair. The training robot is installed under the chair, and one foot is fixed to the training robot footplate with a band. The display in front of the trainee visualizes the force applied at the tip of the toe as a filled yellow circle, the radius of which is related to the amount of the force in the dorsiflexion direction (Figure 5). The trainee is asked to adjust the force so that the filled yellow circle matches a red circle, which displays the amount of target force. The red circle is displayed only when the angular command moves in the plantar flexion direction. Constant eccentric contraction force is achieved with the trainee adjusting the force on the tip of the toe while the robot moves in the plantar flexion direction. The angle provided to the front motor is shown in Figure 6. Initially, the two motors keep the footplate at dorsiflexion position. When both actual angles exceed these angles, the rear motor angular command maintains the current angle. By lifting the footplate, an ankle joint movement range is obtained. Next, the angular command of the front motor begins moving in a sine wave-like signal, as illustrated in Figure 6. The amplitude of this wave is calculated based on the ROM of the subject’s ankle. When the command input signal reaches the sine wave peak, the command input remains constant for a few seconds (point A). Finally, the front motor command falls. At this moment, the subject is still required to maintain the target force value mentioned previously (point B). As a result, training is accomplished with both the preferred strength and speed. It has been already shown in [18] that the reproducibility of the eccentric contraction force improves during the movement with the force bio-feedback loop. It is highly possible that such a training also has a higher clinical effect on the motor control performance of eccentric contraction because of the higher reproducibility of force during training. 

A total of four sets of training was performed with two sets for each foot. Here, one set consists of eight exercises of eccentric contraction with 3 s. One-minute breaks were taken between sets. The target force was 40N at the tip toe. 

### 2.3. Evaluation and Training Period

The subjects in this study consisted of 11 elderly individuals with walking ability at a nursing care center. Subjects with lower limb pain and serious medical histories (central nervous disorders and orthopedic disorders that severely affect walking) were excluded in this study. The contents of the study were explained to the subjects and families prior to the experiment and informed consent was received from all subjects. The present study was carried out with the authorization of the Human Research Ethics Committee of Saitama University. 

The duration of the evaluation and training period took approximately two months. The training sessions were conducted four times within one month, in sets of eight, twice in one session. The interval time was 1 min between each set. In this study, we conducted a cross-over comparison test to determine the training effects of the robot as shown in Figure 7. Subjects were randomly divided into two groups. In the first month, one group conducted training with the training robot. This phase is referred to as the training phase. On the other hand, the other group did not engage in training. This phase is referred to as the control phase. One month later, the two groups were interchanged and the same process was carried out. The first and latter halves were defined as the first intervention period and the last intervention period, respectively. The number of training times was set to four, and evaluation was conducted three times: before and after the first intervention period and after the last intervention period.

The TUG and 5-m walking tests were conducted as methods for the evaluation of training effects with the use of the training equipment. Stopwatches were used for measurements in both tests. The time taker was blinded.

The 5-m walking test is a test for the assessment of walking speed. In a previous study, the 5-m walk test was used for motor-performance measures [19]. This test measures the time it takes to walk a distance of 5 m at a comfortable pace.

The TUG test assesses functional dynamic capacity, and this method has also been shown to correlate with measures such as the Berg Balance Scale [20], gait speed [21], and probability of fall [22]. In performing the TUG test, the subjects are timed while they rise from an armchair, walk at a safe and comfortable pace to a line on the floor 3m away, turn and walk back to the chair, and sit down again [23]. This provides a highly reliable index for musculoskeletal ambulation disorder symptom complex, and has been shown to be strongly linked to everyday functionality, including lower limb strength, balance, walking ability, and fall likelihood [24]. This measurement is widely used for ambulatory function assessment in the elderly. Subjects were allowed to practice once prior to the actual measurement, and measurements were obtained from the two subsequent times. The best time was selected as the test result in order to reduce the uncertainties involved in sampling errors.

The normality of data was assessed by the Bartlett’s test. In case any non-normal data was detected, appropriate non-parametric testing was used. The Kruskal-Wallis test was used for the intergroup comparison of walking speed and TUG time at different times (i.e., before examination, first intervention period, last intervention period). Furthermore, in this study, all the subjects were divided into a training phase and control phase regardless of a cross-over test, and a comparative investigation was performed. The Wilcoxon signed-rank test was used for comparisons between the two groups. The level of significance was set at 5%. Statistical procedures were performed using MATLAB 2016a.

## 3. Results and Discussion

Figure 8 shows the walking speed during the 5-m walking test. The left figure (group A) shows the trend toward higher speed after the first intervention period. On the other hand, group B had higher speed after the last intervention period. However, there was no significant difference between each period.

Figure 9 shows the TUG time. There was no significant difference between each phase.

Figure 10 shows the walking speed and TUG time between the training and control phases among all the subjects together. A significant difference was observed in the 5-m walking test (*p* < 0.05). However, there was no significant difference in the TUG time.

As described above, we conducted a clinical study of a training device designed to assist in the eccentric contraction training of the tibialis anterior muscle. The purpose of this study was to investigate the effects of walking training with the use of the device. The results indicated that there was no significant difference in the walking speed during each period measured using the cross-over test. However, there was a trend toward improvement, and a significant difference was observed between the training and control phases in all subjects.

An advantage of the present study is shown via the eccentric training focusing on muscle contraction mode, not muscle strengthening. Many previous studies recommend high-intensity and high-frequency training for muscle strengthening [25]. The training sessions of the present study were conducted four times within one month, which were conducted in sets of 8, twice in one session. Therefore, it was assumed that muscle strengthening could not be expected. In addition, we used the Functional Reach Test (FRT) [26,27] for our pre-research. The FRT is a dynamic balance task that measures the distance that subjects are able to reach forward while maintaining a fixed base. Forward motion tasks, such as the FRT, have indicated increased co-contraction by increased muscle activity of the tibialis anterior and gastrocnemius muscles. However, our pre-research did not show an increased reach distance. We speculate that the robot training cannot expect muscle strengthening to resist forward movement of the center of pressure. Therefore, the present study did not measure muscle strength. Although, there was an observed difference between the training and control phases in all subjects, as indicated in Figure 10. Actually, the tibialis anterior muscle during gait indicated activity of only about half of the maximal voluntary contractions [7]. Moreover, walking speed and functional performance related to walking ability improved during low-intensity training [8]. Therefore, even with the low strength and frequency of training used in this study, we demonstrated the possibility of using this experimental device to improve walking ability through eccentric contraction training of the tibialis anterior muscle.

There was no significant difference in time in the TUG test. The results in [16] also shows similar tendency. The TUG test consists of a battery of tests that assess complex movements, including standing up and changing direction. The standing up movement includes kinetic tasks that differ from walking, and exhibits characteristic activities that correspond to general muscle activity. Changing direction is affected by a decline in overall physical flexibility. Therefore, it is likely that, in addition to tibialis anterior muscle strength, changing direction requires strength from other muscles, overall joint ROM, and balance.

It is known that the cross-over comparative experimental design may indicate a training effect caused by the carryover effect. However, no improvement was found during the control phase in the group that initially performed the training. The fact suggests that there was no carryover effect and that the results obtained were due to the training itself. The scores for all subjects decreased 1 month after the intervention. The results indicate that the effect of the proposed training is limited to a short-term period and it is important to have continuous training. 

The limitations of this study include the fact that, since the training effect was investigated at only one performance level, we may not have fully investigated walking ability in detail. Improvement in walking ability might cause changes in overall physical dynamics, including the vicinity of the ankles, which are in direct contact with the tibialis anterior muscle. However, in order to investigate this in detail, a larger-scale experimental environment that includes the use of a three-dimensional motion analysis system would be required.

## 4. Conclusions

This study discussed the possibility of using the training system with force bio-feedback to improve walking ability through eccentric contraction training of the tibialis anterior muscle. Evaluation of 11 elderly participants with months training period was conducted through a cross-over comparison test. There was a trend toward improvement on 5 m walking test, and a significant difference was observed between the training and control phases in all subjects. Since we investigated the effect of training in clinical settings, we were unable to perform a detailed analysis of the walking movement. We would like to conduct further studies on the reliability of this experimental device that also address these issues.

## Figures and Tables

**Figure 1 sensors-19-01288-f001:**
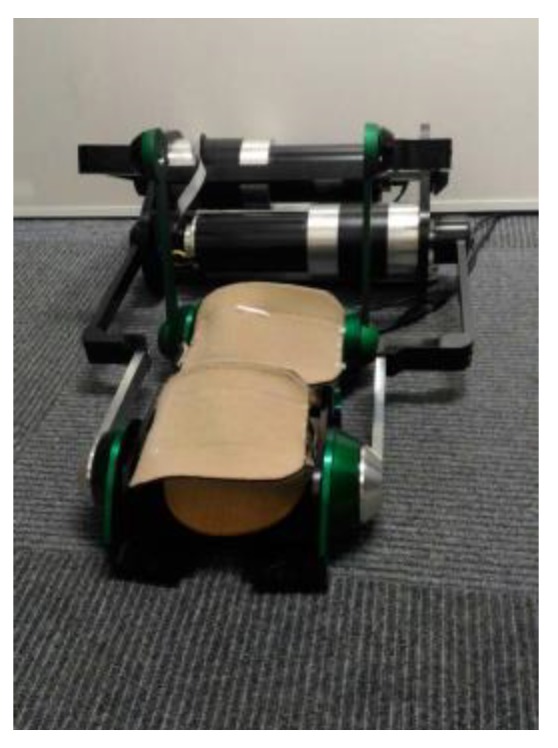
Image of the training robot.

**Figure 2 sensors-19-01288-f002:**
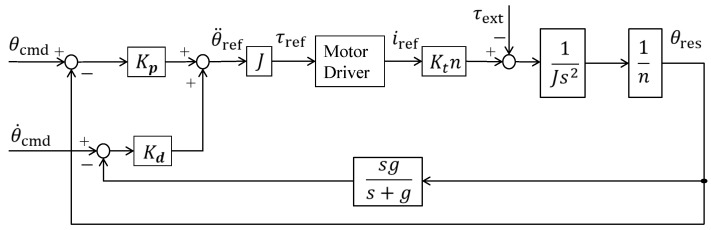
Diagram of the control system.

**Figure 3 sensors-19-01288-f003:**
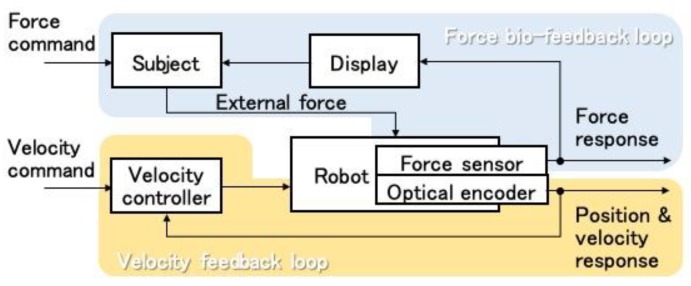
Training scene layout.

**Figure 4 sensors-19-01288-f004:**
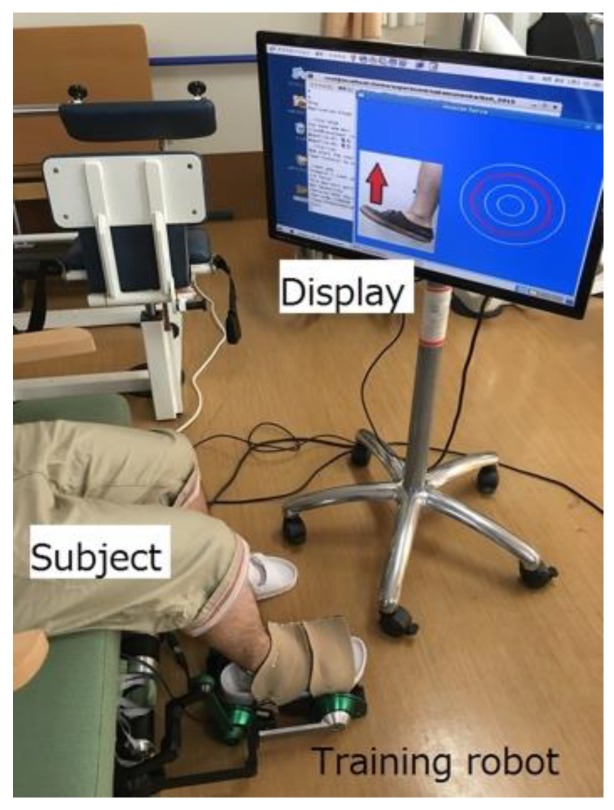
Actual training scene.

**Figure 5 sensors-19-01288-f005:**
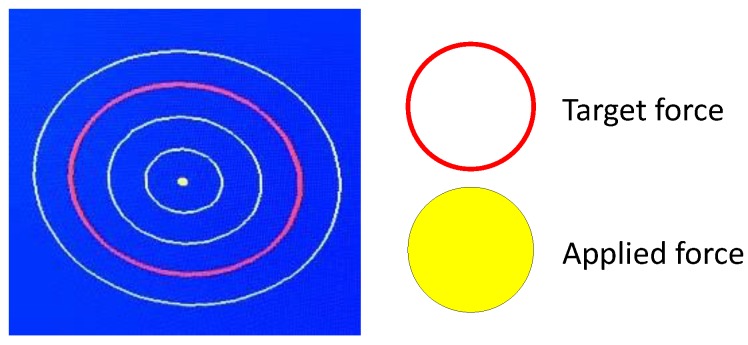
Visual feedback.

**Figure 6 sensors-19-01288-f006:**
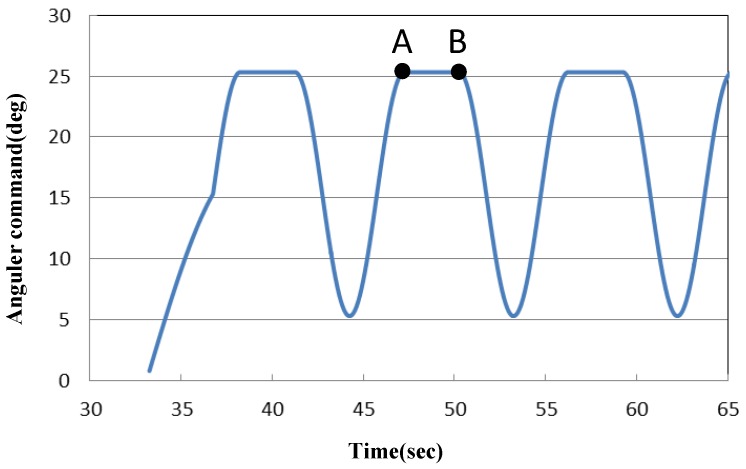
Angular command for the front motor.

**Figure 7 sensors-19-01288-f007:**
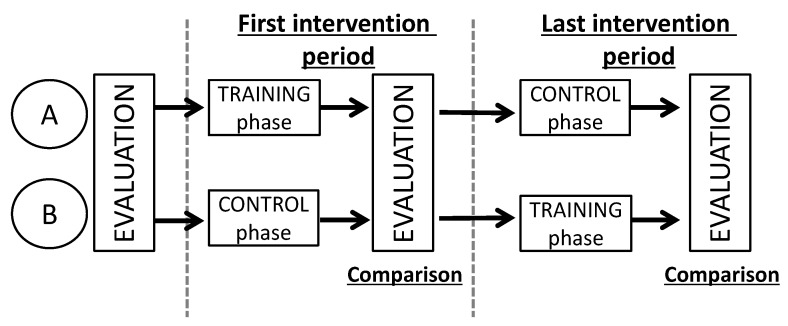
Evaluation and training period.

**Figure 8 sensors-19-01288-f008:**
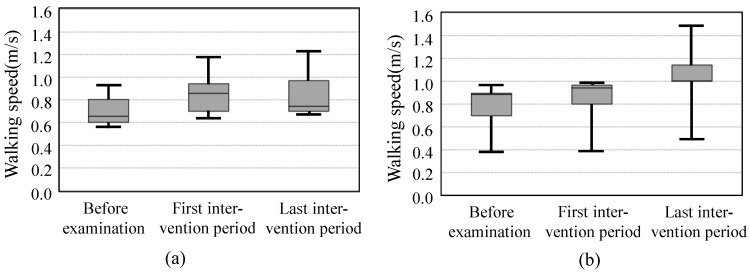
Walking speed during 5-m walking test. (**a**) Group A; (**b**) Group B.

**Figure 9 sensors-19-01288-f009:**
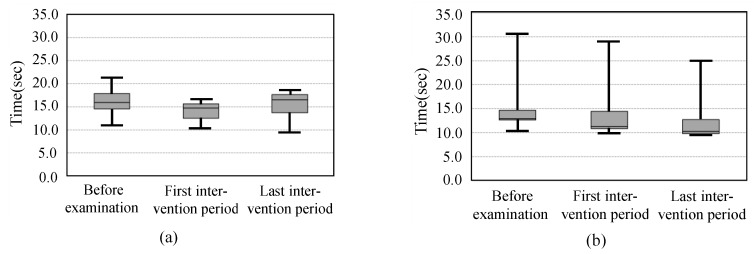
The TUG time. (**a**) Group A; (**b**) Group B.

**Figure 10 sensors-19-01288-f010:**
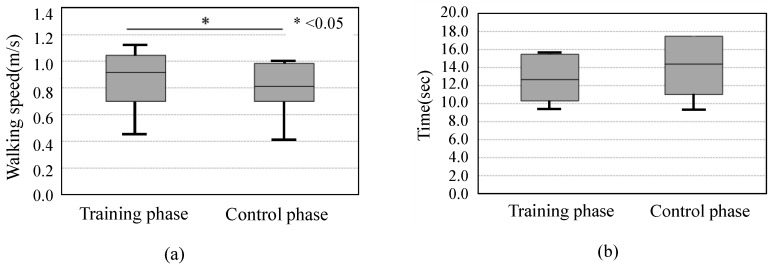
The score of all subjects. (**a**) 5-m walking test; (**b**) TUG test.

**Table 1 sensors-19-01288-t001:** Specifications of the training robot.

Force sensor	**Manufacturer**	**Leptrino Inc.**
	Model number	FFS055YA501U6
	Rated load	F*x*,*y*,*z*-:axis: 500 N
		M*x*,*y*,*z*-:axis: 4 Nm
DC motor	Manufacturer	maxon motor
	Model number	RE 65
	No load speed	3090 rpm
	Max continuous torque	809 mNm
	Rated output	250 W
Planetary Gearhead	Manufacturer	maxon motor
	Model number	GP 81A
	Reduction ratio	25:1
Encoder	Manufacturer	maxon motor
	Model number	HEDL 5540
	Counts per turn	500
	Number of channels	3

**Table 2 sensors-19-01288-t002:** Parameters of the control system.

Parameter	Value
$K_{p}$	407
$K_{d}$	53
J	0.000128 (kg m^2^)
$K_{t}$	0.214 (Nm/A)
n	25
g	62.8 (rad/s)

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
