# Peer review of "Robot-Assisted Eccentric Contraction Training of the Tibialis Anterior Muscle Based on Position and Force Sensing"

_sensors, 2019, doi:10.3390/s19061288_

Round 1
Reviewer 1 Report
Summary:
This paper examined the clinical effects of an ankle exercise device. Overall, the idea appears to be interesting and addresses the eccentric contraction training of the tibialis anterior muscle.
Comments:
The methods and results were clearly described. My only concern is the small number of participants and I have some concerns about the methods.
Regarding the 11 subjects, it seems that the general information was not clear.
How long it took for each session? And what kind of tasks would the participants perform? The authors are suggested to specify better the training exercises and the possible recovery outcomes they are conceiving.
In line 207, the upper figure maybe a typo error.
And compared to the previous study (ref 12), compared to the conclusion of shortening TUG time in the previous study, it may add value about discussions regarding TUG time.
Besides, the robot system proposed visual feedback, while I would suggest the authors to report the evaluation of the feedback and how participants reacted to it?
Author Response
Dear reviewer,
Thank you for your kind comments.
The answers are described in the PDF file.
Best regards

Reviewer 2 Report
The manuscript lacks a description or important section on velocity feedback and force-feedback including poor figure and table quality. I list comments for improvement as:
Abstract Line 22: Please observe the word Conclusion: in the abstract. Revise this sentence.
Page 2 Line 63: “Biodex System4”. Why this number 4 here?
Page 3 Line 87 Table 1: Increase the font in all rows and columns to improve the reading visibility.
Page 3 Line 93 Figure 1: Label figure 1 appropriately.
Page 3 Line 104 Figure 1: Increase the font and line for this Figure too. Define all the parameters shown in Figure either in the caption or in the text.
The manuscript should include one section or sub-section that should include a brief mathematical or theoretical description of how the velocity feedback in the robot controller and force bio-feedback based on force visualization were achieved.
Quality of most figures is poor. Please update (major revise) Figure 8-10 to improve the visibility and quality of the figure. Include legends and larger fonts where necessary and also increase the thickness of the plots.
The discussion section starts with the word “This” without stating what this refers too. Revise the sentence.
Poorly prepared conclusion section. Please revise & improve it.
The revised manuscript must include a theoretical section on velocity feedback and force bio-feedback principle of operation of this device. It should drastically improve the visibility of all the figures and tables, abstract and conclusion section. Without this, the manuscript is not in the in the publishable state.
Author Response

(The authors gave the same response as above.)

Reviewer 3 Report
Authors do not present a scientific contribution or important apportation. The contain of the manuscript is very confusing. The sections of the manuscript have poor technical information about the scientific contribution and advantages of the proposed device. The manuscript does not contain detail description of the operation, design, results and discussions of the proposed device. This work does not add relevant results and discussions. This work needs more experiments and discussions.
Author Response

(The authors gave the same response as above.)

Round 2
Reviewer 1 Report
Most of the comments have been answered. While I would recommend the authors to rephrase their statement of the possible clinical outcome.
Author Response
We appreciate your valuable comments. Please see our response in the attached file.

Reviewer 2 Report
The revised manuscript has improved. The figure sizes are small (see, Fig 3 and 7, for example) but these can be adjusted during the proof-reading stage.
Author Response

(The authors gave the same response as above.)

Reviewer 3 Report
This manuscript requires more modifications to improve its contain. Authors must add the following points:
-Abstract section. This section is confusing. It must include the main contribution or advantages of the presented study. Which clinical effects were considered in this study? Which methods were used? Which results were measured? Which conclusions were obtained?
Introduction section. This section must mention more recent references (e.g., references between 2015 and 2019). This section contains short paragraphs, which must be corrected. This section has weak information. Authors must include more relevant information with data about the main problem and limitations of other researches about the muscle activity during walking. Which are the main advantages of the presented study with respect to others researches?
Materials and methods. This section contains small subsections. This subsections must include more technical information about their contain. For instance, these subsections must add more technical information about the used methodology and descriptions related with the experimental setup as well as statistical analysis.
Results section. This section contains few information. This section must be included with discusion section. The new section of results and discussion must mention more discussions about the obtained results.
Author Response

(The authors gave the same response as above.)
